# A Frame Theory of Energetic Life: A Twisting Energy Solidified on the Holographic Fractal Structure

Yanju Wei [1,*,†], Yajing Yang [2,†], Yajie Zhang [1], Zhiqiang Mu [1] and Fanlu Bu [1]

1   School of Energy and Power Engineering, Xi'an Jiaotong University, Xi'an 710049, China
2   State Key Laboratory for Strength and Vibration of Mechanical Structures, School of Aerospace, Xi'an Jiaotong University, Xi'an 710049, China
*   Correspondence: weiyanju@xjtu.edu.cn
†   These authors contributed equally to this work.

**Abstract:** Life, as the most mysterious and unique phenomenon on the Earth, has confused humans since time began. Why does life exist as it does and how has the diversity of life developed? We, herein, propose a new theory of energetic life, based on existing energy laws, to interpret the evolution and categorization of physical life forms, from microscopic life to macroscopic life. According to this theory, life is a process in which a mass of energy flows and diffuses in the environment. This energy takes DNA as the three-dimensional blueprint, protein as the basic material unit, and fractal network structure as the framework, so as to solidify from energy and form a semi-solid structure. DNA base pairs simultaneously have dual properties as protein pointers and spatial coordinates, and the multi-level self-similar fractal helix structure ultimately guides the formation of different levels of the fractal structure of organisms. This theory organically links the life phenomenon from microscopic to macroscopic levels, from gene, cell and organ to organism, and it provides a new perspective on life, which may inspire biologists to better reveal the mystery of life.

**Keywords:** life theory; energetic life; DNA; 3D blueprint; fractal structure

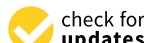



Life, as the most extraordinary phenomenon in the universe, is everywhere on Earth but nowhere else in the observable universe with a radius of around 46.5 billion light years. What is life? How did physical life originate and blossom and how did the diversity of life develop? These questions have plagued humans since the birth of human civilization. Life is a complex, self-sustaining system of chemical reactions [1–5], uniformity and diversity [6]. People cannot even give life a precise definition. Over 100 published definitions of life have been suggested. A popular definition holds that life is a "self-sustaining chemical system capable of Darwinian evolution" [7]. Some scientists have proposed, in recent decades, that a general living systems theory is required to explain the nature of life, instead of examining phenomena by attempting to break things down into components [8].

Several theories have been developed, among which the Gaia hypothesis regards the Earth as a superorganism [9–11]. The Nonfractionability theory takes life as being a self-organizing complex system [12–14]. The Ecosystemic life theory treats environmental fluxes and biological fluxes together as a "reciprocity of influence" [15–17]. The Darwinian dynamics theory argues that the evolution of order in living systems, and certain physical systems, obeys a common fundamental principle from macroscopic order, generated in a simple non-biological system, to short, replicating RNA molecules [18,19]. The operator theory proposes that life is a general term for the presence of the typical closures found in organisms [20–24], and it can also be modeled as a network of inferior negative feedbacks of regulatory mechanisms, subordinated to a superior, positive, feedback formed by the potential of expansion and reproduction [25]. The subject of complex systems biology (CSB) developed to study the emergence of complexity in functional organisms from the viewpoint of dynamic systems theory [26–29]. These prevailing theories have built several models for ecological and biological systems and have provided the major relationships,

from quality to quantity, within the inner systems and outer systems of living organisms. However, no perfect theory has been established, and they all only partially explain life characteristics, morphologies and the operation of organisms regarding certain aspects. Life is definitely an extraordinarily complex matter that encompasses all known disciplines. An eligible revealer of what life is would have to synthesize biology, chemistry, mathematics, engineering, informatics, systemology, etc. Unfortunately, due to the detailed nature of the subject, it is very difficult to share knowledge between disciplines. Thus, the synthesis has not yet appeared and a universal theory is still awaited.

The authors, one being an energy engineer and the other an amateur biologist, herein, propose a simple but universal life hypothesis where the living organism is considered as a bulk of energy, which automatically evolves and solidifies on a series of fractal networks based on a 3D blueprint of chromosomes and genes. Taking energy evolution as a clue, this paper connects the theories of genetics, cytology, biology, biological morphology, ecology and life evolution, and provides inspiration for biologists to consider life phenomena from the perspective of energy.

## 1. The "Big Bang" of Life

To avoid being drawn into the ocean of life's diversity, we, firstly, simplify the life phenomenon by tracing it back to its origins. All plants can be finally traced back to an autotrophic cell, while all animals can be traced back to a heterotrophic cell [29,30]. Both these cells evolved from the same primitive cell with both autotrophic and heterotrophic functions. Thus, this cell is the starting point of the "big bang" of the ecosystem. Now we proceed to pursue the cell's evolution to see how it grew into so many complex types of life.

Before starting the journey, we first note that this study assumes that life is a natural phenomenon, rather than a designed one, which evolved automatically and follows all the energy theorems in nature. The germ of a seed is a miniature plant, it absorbs energy and nutrients firstly from the food store and then from the soil via the root system, while animal embryos first absorb nutrients through the mother and then absorb external energy through the digestive system after birth. These energies and nutrients are converted into the bodies of plants and animals, which do not change in shape, but expand tens of thousands of times in size. Since the nutrients are the carriers of energy, we can say that living organisms are a mass of moving, accumulated energy. Then comes the interesting question: How does energy evolve?

We start from the evolution of organism morphology and structure. Multicellular life always develops from a germ cell and builds up with tremendous increase in the quantity of copied cells with the same DNA. The organism that develops is energy supported, and is an entity determined by DNA coding, with a certain morphology and inner structure. Since the germ cell is spherical, we first show how the morphology of life developed by comparing it with that of an impacting droplet.

## 2. A Droplet of Life

### 2.1. Morphologies of Impacting Droplets

When a water droplet (density, $\rho_d$) impacts a water pool (density, $\rho_p = \rho_d$), it converts to a sinking vortex ring with a tail left behind, as shown in Figure 1a, which is comparable to the umbrella and handle of a mushroom (Figure 1b). Thus, the three-dimensional (3D) structure could be termed a liquid mushroom. It grows and, finally, vanishes due to the mass diffusion and dissipation of the energy brought by the droplet. The liquid mushroom is, in essence, the dissipation structure of the kinetic, gravitational and surface tension energies attached to the droplet acted on by the resistance of the interface and flow resistance forces. The biological mushroom, growing up from the earth, is a dissipation structure of a buoyance energy attached to a bulk gas slightly lighter than the air, but with a much larger time scale. The difference is that the liquid mushroom has a fluid structure, while the biological mushroom solidifies this with hyphae.

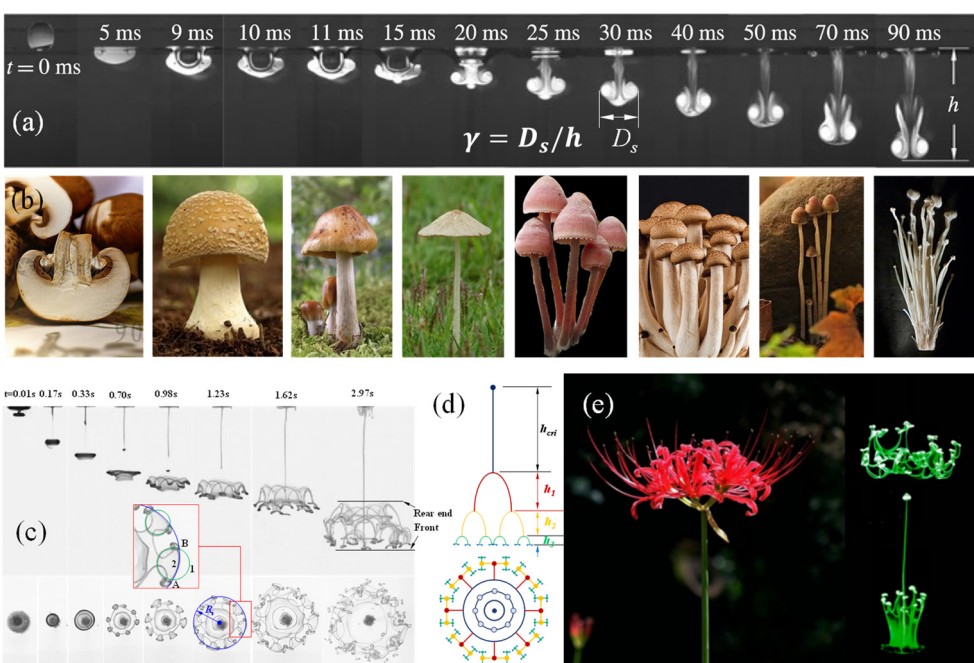

**Figure 1.** (**a**) Comparison of the morphologies of sinking vortex ring [31] "Reprinted with permission from [Abhishek Saha et al.], [The Journal of Fluid Mechanics]; published by [Cambridge University Press], [2019]" with (**b**) mushrooms, (**c**) High speed images showing side and bottom view of the morphological evolution of the vortex ring and (**d**) its abstract structure, (**e**) Morphologies of manjusaka vs. the bifurcation flower [32]. "Reprinted with permission from [Yajie Zhang et al.], [Physica of Fluids]; published by [AIP Publishing], [2021]".

Interestingly, when $\rho_d > \rho_p$, the liquid mushroom soon disintegrates and evolves into a bifurcation flower (Figure 1c–e), via alternating domination of the gravitational acceleration and Rayleigh–Taylor instability, which can be precisely depicted with a set of dimensionless formulae [31,32].

When $\rho_d = \rho_p$, the merging process consists of three stages, namely, rapid penetration, non-monotonic propagation and deceleration, where the inertia, surface wave and viscous drag take turns to dominate the motion. The penetration height h could be described with a set of dimensionless equations:

$$\text{Stage I}: \widetilde{h} = \frac{1}{2}\frac{t}{\tau_i}, \text{ for } 0 < t < \tau_i; \tag{1}$$

$$\text{Stage II}: \widetilde{h} = \frac{1}{2} + \frac{1}{4\pi}\frac{\tau_c}{\tau_i}\sin\left(2\pi\frac{t-\tau_i}{\tau_c}\right) + \widetilde{h}_v\ln\left(\frac{t-\tau_i}{\tau_v}+1\right), \text{ for } \tau_i < t \leq t_0; \tag{2}$$

$$\text{Stage III}: \widetilde{h} = \widetilde{h}_0 + \widetilde{h}_v\ln\left(\frac{t-t_0}{\tau_v}+1\right), \text{ for } t > t_0. \tag{3}$$

where $\widetilde{h} = h/D_d$, and $\tau_i = D_d/U_d$, $\tau_c = \sqrt{\sigma/\rho D_d^3}$, $\tau_v = (4D_d)/(3C_DU_{p,0})$ are the time scales of inertia, capillary and viscous drag stages, respectively. $D_d$ denotes the initial diameter and $U_d$ the impact velocity of the droplet, and $\sigma$ the surface tension of the pool.

The droplet with $\rho_d = \rho_p$, has a vortex ring that ends up following Equation (3). However, when $\rho_d > \rho_p$, the vortex ring disintegrates at the critical height given below:

$$\text{Critical height}: \widetilde{h} = \ln(1+Fr), \text{ at } t = \tau_v \cdot Fr. \tag{4}$$

where the Froude Number $Fr = \sqrt{\frac{3C_D\varepsilon}{4(1-\varepsilon)}} \cdot \frac{U_0}{\sqrt{gD_0}}$ with $C_D$ denoting the flow coefficient, $\varepsilon = \rho_d/\rho_p$ the density ratio and $g$ the gravitational acceleration. The critical height also follows Equation (4).

The "fingers" produced from the disintegration move down and bifurcate at:

$$\text{Criterion}: \ U_a = U_p. \tag{5}$$

where $U_p$ is the penetration velocity and $U_a$ the velocity component caused by the acceleration $a = (1 - \varepsilon)g$.

This set of dimensionless controlling formulae is actually the operating mechanism of the bulk of energy. The evolution of the impacting droplet is highly comparable to that of a plant seed: droplet to seed, tail line to stalk, vortex ring to bud, and bifurcated flower to real flower. The droplet is, indeed, the seed of the bifurcation flower.

The bifurcation flower is a result of an automatic process following the law of motion of objects. From an energy point of view, the bifurcation is a dissipation process of kinetic energies, the gravitational potential and the surface tension carried by the droplet passing across the gas–liquid interface, during which the responses of the environment, such as the resistances of the interface and flow, play a vital role for the growth of the liquid mushroom and bifurcation flower.

These phenomena are enlightening in the following ways: 1. Whatever the morphology of an energy dissipation structure is, it is an outward manifestation of a bulk of energy; 2. Automatic evolution has a very simple nature; 3. The dissipation of a piece of simple energy leads to regular but complex fractal structures; 4. Different development stages of evolution are affected by common external forces, but the dominant force switches in different stages; 5. The dissipative energy structure follows the rules of fluid dynamics.

Obviously, life is completely compliant to the above five characteristics. If the droplet can continuously acquire energy and mass from the outside world, the bifurcation flower may grow into a huge plantlike structure.

### 2.2. Morphologies of Organisms

The liquid mushroom is produced by the directional energy injection of one liquid into another miscible liquid. The embryo of mammals, such as the human embryo, for instance, is full of fluidity and is inflated by the energy supply of the umbilical cord, thus, it has the primary morphology of a vortex ring, although highly ellipsoidal.

However, for plants with solid cytoderms lacking fluidity, the energy has to flow forward to form a column, e.g., trunk, branch, petiole etc. Moreover, at the end of the petiole, the 3D pipe is deformed to a 2D plate, namely the leafage, and the column flow inside the petiole is, thus, switched to plate flow, as in the flattening of the end of a water pipe or two laminar jets colliding, where the outline of the liquid sheet is similar to the shape of broad leaves. Note that we are only considering the basic shape of the organism and not the details and differences between species.

In a word, it is the energy, or fluid flow, that determines the basic morphology or shape of the organism.

### 2.3. Energetic Gene
2.3.1. Gene of the Droplet of Life

Droplets, having certain properties and initial impacting parameters, such as liquid viscosity $\mu$, density $\rho$, surface tension $\sigma$, drop diameter $D$, and impact velocity $U$, etc., undergo exactly the same processes and grow to the same morphologies following the controlling functions. The overall performance of the liquid mushroom could be described as that of the energy indexed as $\mathbb{I}_d = I[\mu_d, \rho_d, \sigma_d, D_d, U_d, h_d, t_1, t_2, t_3, \ldots]$, which evolves in the environment indexed as $\mathbb{I}_0 = I[\mu_0, \rho_0, \sigma_0, D_0, U_0, h_0, \tau_1, \tau_2, \tau_3, \ldots]$, under the mechanism expressed as $f = f[\text{Equation}(1), \text{Eqquation}(2), \text{Equation}(3), \ldots]$.

Note that the coupling of the energy and environment, $\mathbb{I}_d : \mathbb{I}_0$, is complete and exclusive to define the interaction system, and, thus, it is acceptable to define the gene of the liquid mushroom as:

$$G = G\left[\frac{\mu_d}{\mu_0}, \frac{\rho_d}{\rho_0}, \frac{\sigma_d}{\sigma_0}, \frac{D_d}{D_0}, \frac{U_d}{U_0}, \frac{h_d}{h_0}, \frac{t_1}{\tau_1}, \frac{t_2}{\tau_2}, \frac{t_3}{\tau_3}, \cdots\right] \tag{6}$$

To distinguish this from biological DNA, here the gene $G = G[\mathbb{I}_d : \mathbb{I}_0]$ is termed "energetic DNA" ($DNA_E$). Interestingly, G also has a double-stranded structure like biological DNA ($DNA_B$).

While the gene determines the essence of the evolving energy, the mechanism function $f = f(G)$ is the dynamic progress concerning the historic evolution of morphologies. If we substrate the initial energy input ($E_0$) out from $G$, life can be expressed in the form of a function:

$$L_f = E_0 \cdot f(G) \tag{7}$$

where $E_0 = E_k + E_g + E_\sigma$, the subscription $k$ denotes kinetic, $g$ gravity and $\sigma$ the surface tension.

Since the living organism has the same mapping relationship with $DNA_B$ as that of the liquid mushroom with $DNA_E$, $DNA_B$ can be regarded as representing the type of biological energy that evolves into the living organism. Then, the structure, morphology and functions of cells, organs and the body of one living organism are the results of the flow and field of the energy. The cells construct the pathways of energy flow by duplicating new cells with proper shapes and structures concerning the mechanical requirements. The cells make their structures by accumulating proteins, fats and minerals. For instance, the neural cells, with radical neuro-synaptics, can be regarded as a solidified electric charge field.

### 2.3.2. The Essence of Life's Evolution

Once we know that the nature of life is the materialized representation of the bulk of energy $\mathbb{I}_d$ (energetic life) in the environment $\mathbb{I}_0$, the evolution of life is mostly excited by the variation of the environment, since DNA is relatively stable. The morphologies of various organisms are the adaption results of $\mathbb{I}_d$ to $\mathbb{I}_0$. The environment an organism lives in is indexed by $\mathbb{I}_0$, including the systems of astronomy, geography, geology, atmospheric circulation and the water system, and the cooperation and competition of other living creatures, etc.

Thus, several biological extinctions and outbreaks of life in Earth's history reflect tremendous changes and the severe unsteadiness following the new status of the environment at that time. The former lives became disused by the vital change, but the surviving life seeds attempted to acclimatize to the new environment and, thus, evolved into various forms of life. However, most newly born lives die due to environmental oscillation, and the life forms that eventually stabilize do so by adapting to the environment.

Darwin's theory of the evolution of organisms centered on environmental influences determining the same group of ancestors; for example, the different skin colors and physiques of Asian, European, African, American, and Australian aboriginal people. Whales evolved from terrestrial to marine animals. The environment can not only create new forms of life, but also carve the somatotypes of organisms. In this way, the phenomenon of Arabidopsis, the active modification of gene coding to accommodate a new environment, challenges the prevailing paradigm that mutation is a directionless force in evolution [33], and could be explained simply as the mutation of parameters in $\mathbb{I}_0$ in its DNA.

In summary, environment plays a dominant role in the creation and evolution of life. The theory of energetic life unifies the theories of life extinction, outbreak and evolution, and points out the direction of life's evolution, i.e., the direction of energy evolution.

## 3. Biological Life

### *3.1. Biological Fractal Networks*

While the morphology of a multicellular organism is determined by its life energy, its substantial body is constructed by numerous cells with the same DNA, but different shapes and textures. To ensure that each cell gets sufficient energy supply, security, drainage, supervision and control, etc., several fractal net systems of lung trachea, blood, lymph and nerve system etc. develop. The networks have the most efficient and economic structures to complete the corresponding functions, such as the veins of leaves. Literature [34–38] has proven that leaf venation is the economical route for water transportation of any given leaf, and, thus, it can be inferred that venation determines the outline of the shape. However, the theory cannot explain why the venation branches in the way it does.

### 3.1.1. Formation of Main Veins and Skeletons

In fact, the leaf blade is developed from a cluster of primordia. The tremendous work provided by Scarpella E et al. [38], shows that it grows from an area of about $10 \times 10$ cells (encircled in a dashed blue line in inside a hollow to an inflated "balloon". During this process, the leaf vein also develops from a line segment into a network. Obviously, the primary and most important function of the veins is to supply all the cells with nutrients with equal opportunity. Note that water in the mesophyll has a limited diffusion length (R), which could be estimated to be 50~100 μm, about the length of one to two dozen cells. With the growth of the leaf blade, the older veins must grow both axially and radially and differentiate into secondary veins, due to the limit of R. Based on this simple principle, synchronized duplication of all cells leads to a fractal venation of leaf veins. The main skeleton of leaf venation is formed as early as when the leafage length is around 200 μm, an ignorable size compared to the adult leaf, much as the animal embryo develops from an ignorable size into the adult body.

Obviously, the arabidopsis cells divide isotropically, which leads to a round leaf blade. What happens if the cells proliferate anisotropically?

Figure 2 shows the formation route of various leaf outlines based on the anisotropic division of leaf primordia. Given a cluster of leaf primordia with the typical length ~R, the cells on the symmetric axis must differentiate into vein cells and, thus, form the first vein to transport water and minerals to support the division of the germ cells. This vein then develops into the main vein of pinnate and palmate leaves. Since water in the leaf blade is transported via capillary force between the cell walls, the shape of the leaf cells plays a vital role for the leaf shape.

For fibrous-celled leaves, the veins follow the length direction. If all the fibrous cells originate from the leaf base, the leaf displays a radical division, which leads to radial veins, such as in the palm leaf. If the cells are rooted on the main vein, the leaf would express an axial division, resulting in an oblong leaf, such as a banana leaf. Four steps of division are enough to form the basic venation of leaves, as seen in the four level veins of the banana leaf in the inserted Figure 2i.

For rectangular or hexagonal celled leaves, division direction also plays an important role. For instance, alternating the sequences of radial and axial divisions leads to palmate leaf venation, such as in plane tree leaves.

Based on the presumption that water or energy is supplied equally to each cell, but with apical dominance, only the apex cell can divide, while the others grow at the same speed. The apical cell divides only when it grows to the division length ($L_d$), supposing for the $n$th time, then the older cell produced from the $i$th division grows to the length $l_i = 2^{n-i} L_d$, as in the inserted Figure 2ii for apical dominance. This leads to obovate leaves, such as magnolia denudate leaves. However, if the energy supply to older cells is gradually reduced, it leads to deltoid leaves, such as white poplar leaves.

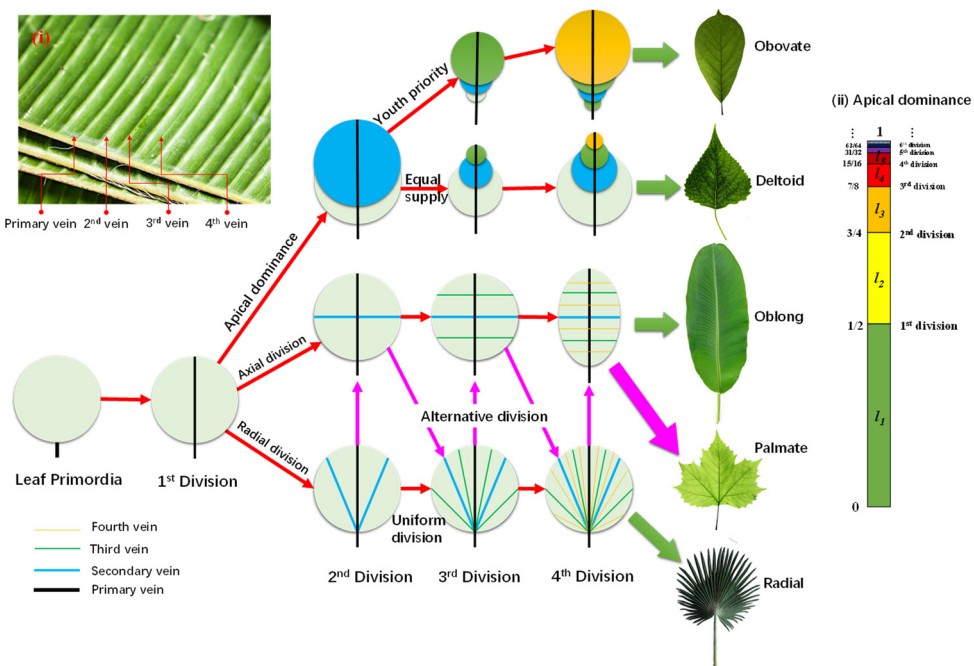

**Figure 2.** Universal formation routes of different leaf shapes, based on different division sequences of leaf primordia.

The main skeleton of animals could be formed similarly from a few steps of primordial division. Take the formation of the human skeleton, for instance, as shown in Figure 3, which primarily formed within five steps of bone primordial division, although the hand and foot bones needed six more steps.

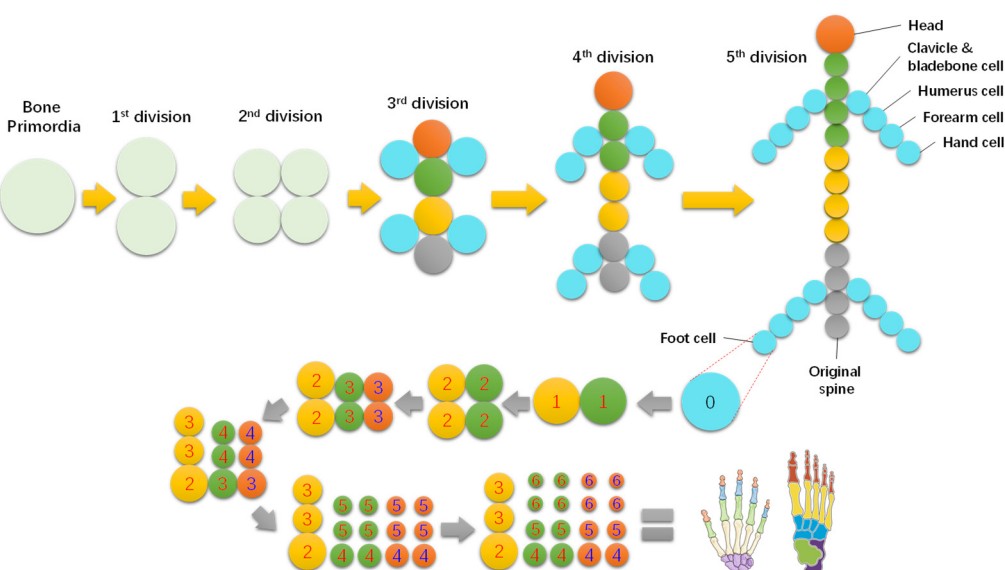

**Figure 3.** Brief formation route of human skeleton and hand and foot bones.

### 3.1.2. Formation of Microveins

Since the main veins of leaves or animal skeletons formed in the first cell division steps, microveins formed in the expansion of leafage or body size, attributed to the duplication and growth of mesophyll/osteogenic cells. New veins are self-similar fractal structures rooted to the primary vein with open ends, as shown in Figure 4a.

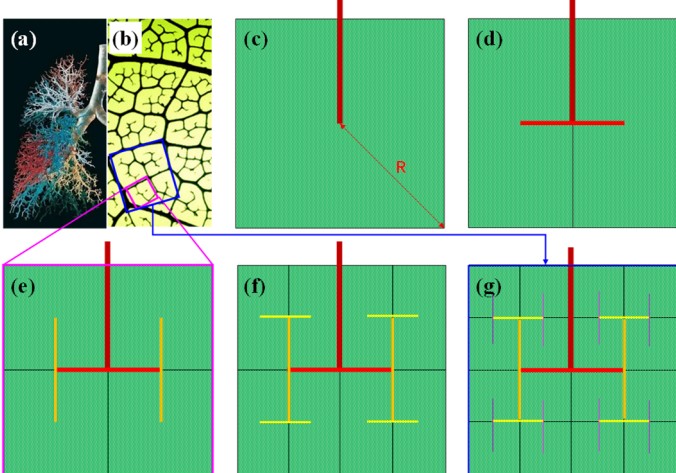

**Figure 4.** Photograph of (**a**) lung trachea and (**b**) leaf microveins. (**c**–**g**) the formation diagram of typical micro leaf veins/bronchus based on cell region partitioning. Note that the side length of the largest square is doubled in each step from (**c**) to (**g**).

The formation rule of the fractal network is simple. It is based on the following three facts: (1) The maximum diffuse distance (R) of water among mesophyll cells is up to one or two dozen times the cell length, around 0.2~0.4 mm; (2) Cell regions are divided along the veins; (3) Water can only flow out of the veins at the dead end, as the vein wall is watertight. Figure 4b–f shows the brief generation steps of microveins. For a given primary square area (PSA) with the maximum diffusion length R (Figure 4b), the primary vein (dark red line segment) ends at the center. With the division and growth of mesophyll cells, the PSA are split in the horizontal direction into two regions along the primary vein, then two first level subveins (red line segment) extend from the end of the primary vein into the center of the two regions (Figure 4c).

Then, the left and right regions continue to split simultaneously in the vertical direction, leading to two up and down regions, followed by the extension of second level subveins from the end of the first level subvein to the center of two new regions (Figure 4d–f). Then, the new regions continue to split alternately in both horizontal and vertical directions, which finally leads to a bifurcation tree of infinite fractal structure. Note that the black lines inside the green regions are also the veins of various levels, but they are weakened to highlight the fractal structure of the colored veins.

### 3.1.3. Fractal Network in Organisms

The fractal network, based on the duplication of cells, is a universal structure in life. As long as a system is generated from a limited number of primary cells, it displays a 3D or 2D fractal network structure, such as the leaf venation, tree trunk, plant root, lung trachea, blood system, nervous system, lymphatic system, and even meat texture, etc., The white connective tissue membrane between muscle cells is actually the fractal structure of periosteum, similar to the capillary vesicles to the heart.

Interestingly, the skeleton is also a two-level fractal structure of one to five branches. Each piece of bone originated from a primary bone cell, and divided from its mother cell. Taking hand bones, shown in Figure 3, as an example, they can be formed in six steps of division from the initial hand bone cell, alternating in radial and axial directions. Each of the sixth level primary bone cell develops into a phalange via structural differentiation, while the formation of the main skeleton is termed functional differentiation.

The process is also simple, as shown in Figure 5. The bone primordia evolve into a phalange via a six-step division with the cells accumulating in the style of a concentric circle. Note that the phalange has a similar hierarchical structure to the cell. The cell membrane develops into the periosteum, the cytoplasm into bone structures, and the nuclei into bone marrow tissue, etc. So, bone marrow contains hematopoietic stem cells.

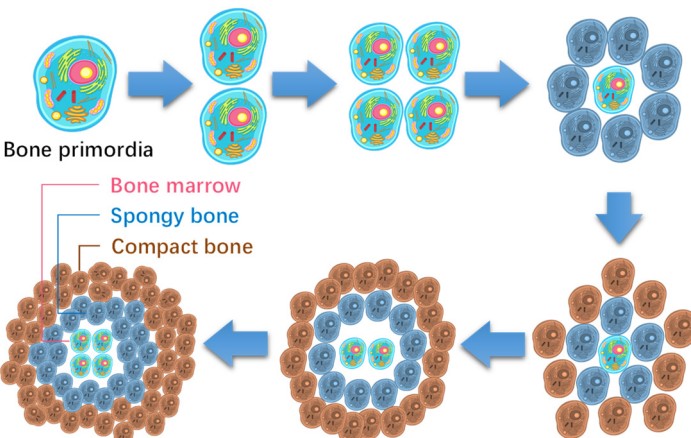

**Figure 5.** Formation process of hand bones from a primary cell within six steps of division.

### 3.2. The Holographic Structure of Life

We have talked about the self-similarities of morphologies and the inside networks of organisms, which are all automatically formed following a simple rule of energy evolution, so that just one primary cell evolves into an organism. The formation of the primary cell should follow the same rule. Since the fractal structure is everywhere in organisms, the similarity should exist not only on the levels of the same category, e.g., network, but also on different levels of genes, cell, organs and organisms. Figure 6 shows the similarities, in cross sectional view, of a cell with that of medial thigh, for instance. Most cell components find their corresponding structure in the medial thigh, such as membrane vs. skin and chromosome vs. marrow, as mentioned above. An organ can be considered a supercell that replaces each molecule of the primary cell with a cell, sharing similar structures but having different basic units, i.e., molecules and cells.

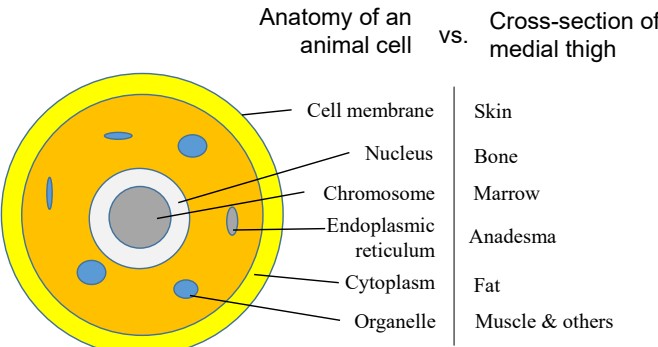

**Figure 6.** Analogy between the anatomy of an animal cell and that of the cross-section of the medial thigh.

Similarities between different levels can be visualized a spiral of genes. The gene strands are folded up through six-level helices to form chromosomes, with the base pairs being the primary units of the DNA helix, and the latter the unit of histone, and so on to the solenoid, the chromatin, the chromosomal microstrip, and, finally, the chromosome. This is similar to the progressive relationship between cells, tissues, organs and organisms. This sounds reasonable, but how are these processes realized?

As known to all, genetic code is the encoded information organized on the double strand of DNA with base pairs as the carrier. It concerns the transmission rule of genetic information, which translates DNA, or mRNA, sequences into amino acid sequences of proteins with three nucleotides as a group of "codons" for protein synthesis.

The gene expression of all organisms has strict regularity; that is, temporally and spatially specific. Temporal specificity means that gene expression occurs in a certain

chronological order. Spatial specificity refers to the different expression of the same gene in different tissues and organs in a specific growth and development stage of a multicellular organism. The more advanced the biological species, the more complex and refined the gene expression rules, which is due to the needs of biological evolution. Although we know its general operating rules, the details of its operation are still unclear.

We herein propose a reasonable functioning mechanism for the operating rules. It is well known that the genetic code is actually a string of protein index. What if we endow the index with a new connotation that of spatial coordinate? Think about three adjacent codons forming a special coordinate value P[C1, C2, C3], which directly equips each position with a certain protein, for example, protein P2@P[C1, C2, C3], P3@P[C2, C3, C4], and so on. Consider the six-level hierarchical helical structure, each protein could also be positioned with $P_i[N_1, N_2, \ldots, N_6]$, where $N_i$ denotes the $i$th level of helix. This promises the continuities of both the spatial position and the type distribution of protein molecules.

From this point of view, the chromosome becomes the 3D fractal blue print of the organism. Since chromosomes have different levels of spiral structures, the organs and cells perform the functions of different levels.

Interestingly, if we combine the spiral structure of the 3D blueprint of life and the law of apex dominance of growth, we obtain a universal progressive spiral of biological structures, as shown in Figure 7, which provides a rational explanation for the morphologies of most plants.

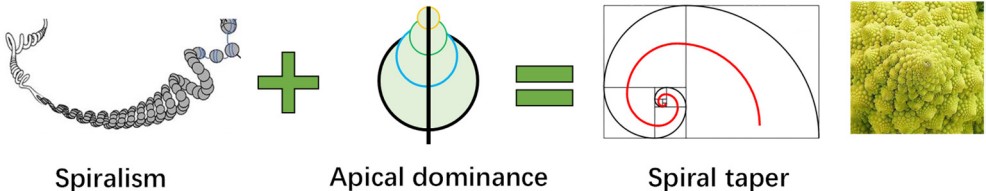

Spiralism        Apical dominance        Spiral taper

**Figure 7.** Formation mechanism of spiral taper and the expressions on various organisms.

## 4. A Conjecture on DNA

So far, we have systematically explained life above the level of genes. The ultimate question is: How if the gene formed? What is the force that holds the base pairs together to form the helical structure and fold step by step into chromosomes? What type of the energy does this structure carry? Is it like the waves on a rope? No one knows. However, as long as we accept that genes are naturally formed, rather than specially designed, they are the fruits of energy flow and evolution.

Note that lower forms of life generally have shorter total DNA lengths, which infers that the original DNA may be a pair of proteins carrying, or involved in, a twisting energy flow. We do not know what type of this energy is, maybe a pair of twisted quanta, but we are certain that the DNA acts like an energy and mass reaper, converting energy and mass into organisms, lives, and even society, with DNA itself and, further, with cells as the basic bricks. In this way, DNA, or the underlying energy, achieves self-proliferation and eternity.

## 5. Frame Theory of Energetic Life

Now we can draw a clear portrait for the growth of organisms and evolution of life from the perspective of energy evolution and hierarchical fractal structure. As depicted in Figure 8, the base pairs carry a bulk of twisting energy accumulates, and they extend in length and fold up into a chromosome with a hierarchical fractal helix. This structure is holographic and serves as a blueprint to direct the construction of cells, and from cells to tissues, organs, and, finally, level by level into organisms.

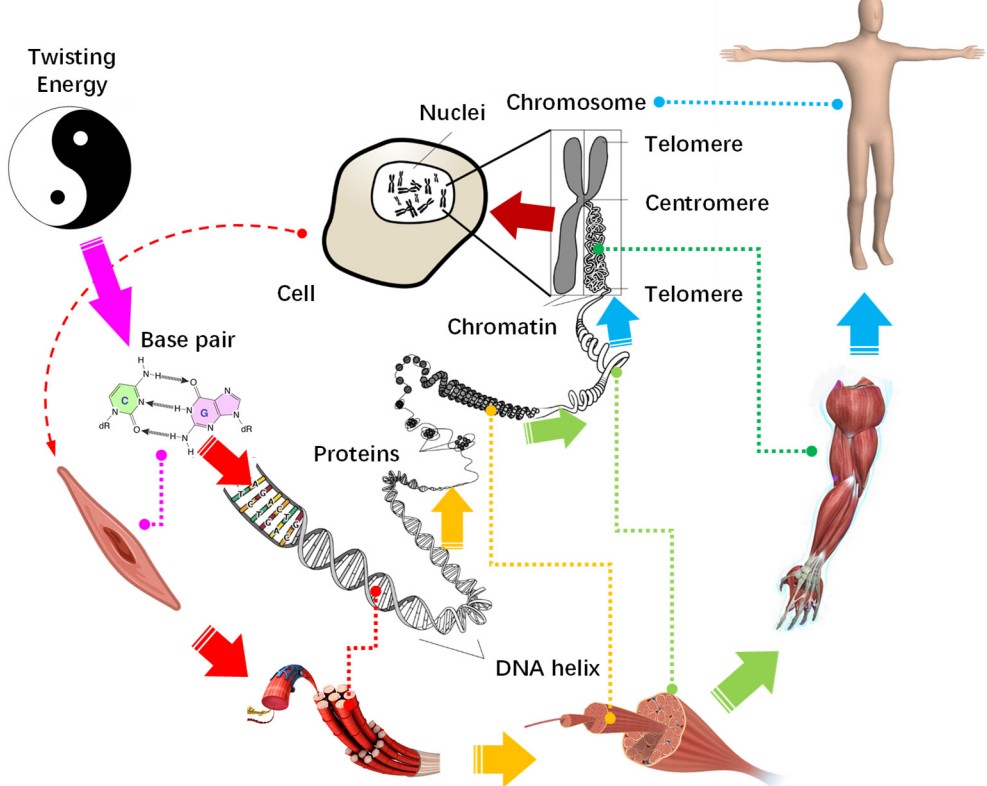

**Figure 8.** The construction route of life from a twisting energy level by level to an organism.

During construction, the germ cell divides into a limited number of tissue stem cells in a few steps, to form the basic structure of the organism, like a sphere transforming into an X-shaped chromosome. After that, each stem cell replicates and differentiates into organs similar to its own hierarchy, analogous to the refinement of the chromosome arms all the way down to base pairs, where the status of base pair to chromosome is the same as that of tissue cell to organs. Since all cells are generated from a primitive cell by binary division, they are all organized by a Y-shaped fractal structure. In this way, with the gene as the reaper, the twisting energy accumulates and solidifies on cell-based organisms. Finally, the energy flows and the organism gains vitality, a life is born.

Here, we have to address that this frame theory of life is rather coarse without enough precise argument, and it is mostly based on logical reasoning and even bold conjecture. More importantly, this theory only gives an explanation for the tangible body, and does address invisible energy, like life and death. However, it still provides a new perspective towards life and may inspire biologists to better reveal the mystery of life.

**Author Contributions:** Conceptualization, Y.W. and Y.Y.; methodology, Y.W.; software, Y.Y.; validation, Y.Z., Z.M. and F.B.; formal analysis, Y.W.; investigation, Y.Y.; resources, Y.W.; data curation, Y.Y.; writing—original draft preparation, Y.W.; project administration, Y.W.; funding acquisition, Y.W. All authors have read and agreed to the published version of the manuscript.

**Funding:** This research was funded by the National Natural Science Foundation of China (grant number 52176128).

**Institutional Review Board Statement:** Not applicable.

**Informed Consent Statement:** Not applicable.

**Data Availability Statement:** Not applicable.

**Acknowledgments:** The authors acknowledge the comments and discussion of the anonymous reviewer, and the suggestions of Baojun Zhang in the School of Basic Medical Sciences, Xi'an Jiaotong University.

**Conflicts of Interest:** The authors declare no conflict of interest.

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
