# Peer review of "A Frame Theory of Energetic Life: A Twisting Energy Solidified on the Holographic Fractal Structure"

_applsci, doi:10.3390/app122110930_

Round 1

Reviewer 1 Report

This is an interesting paper, but I have a few comments and reservations, and therefore cannot recommend acceptance in its present form.

1. Maybe I’m missing something, but this paper – introduced as a theory of the development of life – is only an explanation for the morphology and structure of living things. This, of course, is important in itself. But I think the authors confuse the issue. A spherical droplet is not a living organism. And that is the point – how do we bridge this gap? I fail to see where the authors have contributed here.

2. Perhaps the authors should rebrand the paper in this manner, as an explanation for the morphology and structure of living things. If so, I would suggest they read and reference the following paper by Avshalom Elitzur, “Constancy, uniformity and symmetry of living systems: the computational functions of morphological invariance”, in BioSysems 43, 41-53 (1997).

3. If the authors prefer to address the larger issue of how life developed and diversified, there are many more sources that can be referenced. For example, see Astrobiology Volume 10 (2010), a “Special Collection of Essays: What is Life?” Or C. Mayer, “Life in The Context of Order and Complexity”, in Life 10, 5 (2020), and other references therein.

4. But then there remains that (so far) unbridgeable gap between complex structures and organisms that are actually alive. I fail to see how the authors even address this issue.

Author Response

Reply to Reviewer #1

We thank the referee for his/her comments. We have now addressed the issues raised and correspondingly updated the manuscript. Below are the point-by-point responses (in blue) to this referee’s comments (in black).

Comments of Reviewer #1:

This is an interesting paper, but I have a few comments and reservations, and therefore cannot recommend acceptance in its present form.

Response: Thanks for your recognition and affirmation on this work! We have refined the manuscript according to your comments below.

  1. Maybe I’m missing something, but this paper – introduced as a theory of the development of life – is only an explanation for the morphology and structure of living things. This, of course, is important in itself. But I think the authors confuse the issue. A spherical droplet is not a living organism. And that is the point – how do we bridge this gap? I fail to see where the authors have contributed here.

Response: A droplet is indeed not a life, however, note that the bifurcation flower is a result of automatic process following the law of motion of objects. From the energy point of view, this bifurcation is a dissipation process of the energies of kinetic, the gravitational potential and the surface tension carried by the droplet passing across the gas-liquid interface, during which the re-sponses of the environment, say the resistances of the interface and flow, play a fatal role for the growth of liquid mushroom and bifurcation flower.

These phenomenon give us many enlightenments. 1. The natural process of an au-tomatic evolution has a very simple nature; 2. The dissipation of a piece of simple energy leads to regular but complex fractal structure; 3. Different development stages of evolu-tion are affected by the common external forces, but the dominant force switches in dif-ferent stages; 4. The dissipative energy structure follows the rules of the fluid dynamics.

Obviously, life is completely compliant to the above 4 characteristics. If the droplet can continuously acquire energy and mass from the outside world, the bifurcation flower may grow into a huge plantlike structure.

We have add these paragraphs in the context to bridge the gap of droplet and living organism.

  1. Perhaps the authors should rebrand the paper in this manner, as an explanation for the morphology and structure of living things. If so, I would suggest they read and reference the following paper by Avshalom Elitzur, “Constancy, uniformity and symmetry of living systems: the computational functions of morphological invariance”, in BioSysems 43, 41-53 (1997).

Response: This question is the same one as the next question, and we will reply together in the next question.

  1. If the authors prefer to address the larger issue of how life developed and diversified, there are many more sources that can be referenced. For example, see Astrobiology Volume 10 (2010), a “Special Collection of Essays: What is Life?” Or C. Mayer, “Life in The Context of Order and Complexity”, in Life 10, 5 (2020), and other references therein.

Response: Thanks for your advice. Life is an extremely complex system that almost involves all disciplines of today, scientists have been drawn in the ocean of life complexity and diversity that far beyond the understanding of human ability. We have cited more references to state this complexity that we do not plan to analyze further, since we prefer to avoid this complexity and propose a universal theory from the perspective of energy flow and evolution.

The explanation for the morphology and structure of living things is just a section of this theory, we still prefer to depict the “large issue” how life is originated, developed and diversified. We address that this frame theory of life is rather coarse without enough precise argument, it is mostly based on logical reasoning and even bold conjecture. And more importantly, this theory can only give an explanation for the tangible body, it does nothing about invisible energy, like live and death. But it still provides a new perspective towards life and which may inspires biologist to better reveal the mystery of life.

  1. But then there remains that (so far) unbridgeable gap between complex structures and organisms that are actually alive. I fail to see how the authors even address this issue.

Response: Thanks for your advice. We have add the following content in Section 2.

Noting that the bifurcation flower is a result of automatic process following the law of motion of objects. From the energy point of view, this bifurcation is a dissipation process of the energies of kinetic, the gravitational potential and the surface tension carried by the droplet passing across the gas-liquid interface, during which the responses of the environment, say the resistances of the interface and flow, play a fatal role for the growth of liquid mushroom and bifurcation flower.

These phenomenon give us many enlightenments. 1. The natural process of an automatic evolution has a very simple nature; 2. The dissipation of a piece of simple energy leads to regular but complex fractal structure; 3. Different development stages of evolution are affected by the common external forces, but the dominant force switches in different stages; 4. The dissipative energy structure follows the rules of the fluid dynamics.

Obviously, life is completely compliant to the above 4 characteristics. If the droplet can continuously acquire energy and mass from the outside world, the bifurcation flower may grow into a huge plantlike structure.

--End of reply--

Reviewer 2 Report

The paper attempts to develop a new Theory based on the Energy life casting a new perspective on the evolutionary process of lifeforms. However, the authors bring very few sound scientific arguments to sustain it. The study mostly reiterates some old concepts related to fractal geometry of nature and basic ontogeny knowledge, assuming that a similar process must have occur either if it is about the microscopic or the macroscopic levels of life (lines 335 -336:This theory is able to uniformly explain life phenomena from microscopic to macroscopic under the premise of following the law of energy). The authors resemble the energy holding DNA strands together (lines: 319 - 320: "as to a pair of twisted energies of unknown type, which are related to the string theory and the quantum theory"), when in fact, there are already known the  chemical and physical forces/mechanisms/energies implied in this process. Please, explain in few words what would be the theoretic or practical applications of your theory.

Author Response

Reply to Reviewer #2

We thank the referee for his/her comments. We have now addressed the issues raised and correspondingly updated the manuscript. Below are the point-by-point responses (in blue) to this referee’s comments (in black).

Comments of Reviewer #2:

The paper attempts to develop a new Theory based on the Energy life casting a new perspective on the evolutionary process of lifeforms. However, the authors bring very few sound scientific arguments to sustain it. The study mostly reiterates some old concepts related to fractal geometry of nature and basic ontogeny knowledge, assuming that a similar process must have occur either if it is about the microscopic or the macroscopic levels of life (lines 335 -336: This theory is able to uniformly explain life phenomena from microscopic to macroscopic under the premise of following the law of energy).

Response: Thanks for your question. This frame theory of life is rather coarse without enough precise argument, it is mostly based on logical reasoning of existing concepts and knowledge, and even bold conjecture. This work series up all the disciplines with energy as the main line, taking life as a result of energy flow, we think this will greatly simplify the life research. However, since life involves too much disciplines and most of the life phenomenon are still unknown and most details even not ever seen, it will be a huge project to prove it. But as long as we admit that life is naturally developed rather than designed, energy must play a critical rule in life construction.

Taking turbulence for instance, the detailed structure is also extremely complex that no one now can precisely describe it, but we know the reason for it. It is the result of fluids flowing at high speed, where inertia, momentum and friction controls the status inside the flow. Thus scientist can approach to the truth from macroscopic to microscopic researches. On contrary, life research is on the opposite direction, from microscopic to macroscopic, this is of difficulties magnitudes larger. So, although rough, this frame theory still provides a new perspective towards life and which may inspires real biologist to run on the other way to reveal the mystery of life.

The authors resemble the energy holding DNA strands together (lines: 319 - 320: "as to a pair of twisted energies of unknown type, which are related to the string theory and the quantum theory")when in fact, there are already known the chemical and physical forces/mechanisms/energies implied in this process.

Response: Thanks for your question. In deed, biologist have revealed the truth on the molecular level, all the DNA molecules are connected by chemical bonds. However, what we want to say is what energy does this form of connection carry, like the waves and vibration on strings of a piano, the bonds of steel atoms cannot explain the sound and why it vibrates.

Please, explain in few words what would be the theoretic or practical applications of your theory.

Response: Thanks for your question. The underlying philosophy principle of this work is that everything has its origination, 1 is the mother of N, when it evolves, it obeys one universal rule. So, an idea or thought may develops into a group, an enterprise, and even a country, the larger the organization, the more complex the system. Take medicine for instance, this theory is actually the underlying philosophy of Chinese medicine. If we realize that all the cells and tissues are developed from one single cell, they are the results of topological transformation of the germ cell, and they must have a lot in common. All the cell membranes could be regarded as one huge porous film, they are connected, thus we can apply drugs on the skin to treat internal mucosal disease. Furthermore, massaging foot to treat headache, patting the elbow socket to relieve the stomach pain, and other innumerous treatments. The nerve system could be regarded as a huge nerve cells, the heat and vascular system a huge heart cell, all blood cells one big blood cell, etc. In this viewpoint, a lot of systematic diseases could be treated in new methods.

--End of Reply--

Round 2

Reviewer 1 Report

Reference 7 is from 2010, not 2020.

Author Response

We thank the referee for his/her comments. We have now addressed the issues raised and correspondingly updated the manuscript. Below are the point-by-point responses (in red) to this referee’s comments (in black).

Point 1: Reference 7 is from 2010, not 2020.

Response 1: Thanks for pointing out, we have corrected that.

--End of Reply--

Reviewer 2 Report

I would like you to try a small discussion on similar concepts and compare them with your results/ideas.

See details in the attached file

Author Response

Dear Reviewer, 

Please see the detailed reply in the attached word file.

Regards,

Yanju Wei
